# Influence of the Zeolite ZSM-22 Precursor on a UF-PES Selective Substrate Layer for Salts Rejection

**DOI:** 10.3390/membranes12060553

**Published:** 2022-05-26

**Authors:** Nyiko M. Chauke, Richard M. Moutloali, James Ramontja

**Affiliations:** 1Department of Chemical Sciences, Faculty of Science, University of Johannesburg, Doornfontein, Johannesburg 2028, South Africa; nyikomau@gmail.com; 2DSI/MINTEK Nanotechnology Innovation Centre-Water Research Node, University of Johannesburg, Doornfontein, Johannesburg 2028, South Africa; 3Institute for Nanotechnology and Water Sustainability, College of Science, Engineering and Technology, University of South Africa, Private Bag X6, Florida, Johannesburg 1710, South Africa

**Keywords:** zeolite ZSM-22 materials, polymeric membranes, membrane preparation, nanofiltration membrane, membrane separation, nanocomposite membranes

## Abstract

Fabrication of the ZSM-22/Polyethersulfone (ZSM-22/PES) membranes as selective salt filters represent a growing membrane technological area in separation with the potential of high economic reward based on its low energy requirements. The incorporation of ZSM-22 zeolite material as additives into the PES polymer matrix has the prospective advantage of combining both the zeolite and polymer features while overcoming the limitations associated with both materials. This work investigated the influence of the nature of the silica precursor on ZSM-22 zeolite hydrothermally synthesised using colloidal (C60) and fumed (C60) silica to Si/Al of 60. The successful synthesis of the highly crystalline zeolitic materials was confirmed through XRD, FTIR, and SEM with EDX. The ZSM-22 additives were directly dispersed into a PES polymeric matrix to form a casting solution for the preparation of the ZSM-22/PES selective substrate layers via a phase inversion method for salts rejection. The polymeric PES was selected as an organic network in which the content of the ZSM-22 zeolite (ranging between 0 and 1.0 wt.%), was obtained and characterised by XRD, FTIR, and SEM analysis, as well as water contact angle (WCA) measurement and dead-end filtration cell. The phase inversion preparation method has induced the resulting ZSM-22/PES NF substrates anisotropy, as attributed to a high water flux to the above 700 L·m^−2^·h^−1^; high selectivity and rejection of salts to above 80% is revealed by the obtained results. The materials also exhibited improved antifouling behavior to above 70% flux recovery ratios. As such, the nature of the silica precursor influences ZSM-22 zeolite synthesis as a potential additive in the PES polymer matrix and led to the enhanced performance of the pure PES ultrafiltration membrane.

## 1. Introduction

The search for effective saltwater fillers and selective nanofiltration (NF) membranes is still growing in interest [1,2]. This is surged by the fast increase in population growth, industrial development, climate change, and other related aspects, which expand the requirement for freshwater and subsequently the need for brackish and saltwater desalination via membrane technologies [3,4,5,6]. As reported, saltwater desalination through semipermeable membranes, viz., NF and the reverse osmosis (RO) process, has a promising and efficient production (with about >50% removal efficiency) of portable fresh and clean drinking water [7,8,9,10]. However, these membranes suffer from some limitations. The major limitation in the membrane system, which affects the life performance of the membranes to lower than 10%, is surface fouling [11,12,13].

Surface fouling is the accumulation of suspended solids and microbes onto the membrane’s top surface during the separation process, which results in lowering membrane effectiveness and functionality [14,15]. This then affects the rate of water flux, salt rejection, and permeability of the membranes, which are the key properties of membrane performance. Although this limitation can be easily overcome by pressurising the systems, this, however, increases energy consumption, flushing time, and cleaning steps, which further expand membrane challenges. This also results in shorter membrane life spans and significant economic impacts [12,16]. As such, relatively lower cost, energy requirement, and footprints in membrane processes such as forward osmosis (FO), NF, and RO are required and could render them attractive for desalination approaches. It is well known that the hydrophilicity of the membrane is less susceptible to fouling in a water system [17,18,19]. This is probably a property owning to water molecules’ ability to form a highly organized structural water layer on top of the membrane surface. This, then, inhibits the interactional forces between the surface and the foulant [20,21,22]. Besides, another aspect is the electrostatic interactions between the membrane surface and foulant particles, which increases membrane performance. In principle, suppression of the electrostatic interactions can hamper foulant adhesion on the membrane surface.

In consideration of the water challenges, this study further aimed at fabricating zeolite-polyethersulfone as a NF membrane that exhibits selective salt filtration based on the direct incorporation of different amounts of ZSM-22 zeolite nanoadditives. To the best of our understanding, ZSM-22/PES NF membranes have not been explored in detail by directly incorporating different amounts of zeolites to exhibit superior selectivity in a desalination application, which can overcome serious challenges associated with water scarcity.

In our previous study, ZSM-22 zeolite synthesised using Tetraethylorthosilicate (TEOS), was used in the development of ZSM-22/Polyethersulfone membranes [23]. It was found that this material exhibited a negatively charged framework, well-defined structure, and high crystallinity upon synthesis, which was effective in the development of the resulting NF membranes. However, an understanding of the influence of nature of silica precursor has not been detailed, explored, and established. The nature of the silica precursor seems to have an impact on the crystallinity and retention of the negatively-charged framework in zeolite [24,25,26,27]. As such, this study seeks to explore how the nature of silica precursors in ZSM-22 synthesis will influence the performance of the resulting membrane in salt rejections. 

The ZSM-22 zeolite synthesis is commonly influenced by the nature of the silica precursor, pH of the solution, temperature, and time [28,29,30]. As such, prior optimisation of the zeolite synthesis conditions is paramount [31,32,33]. Here, ZSM-22 as nanoadditive fillers was chosen based on its limited work on membrane fabrication and owing to its well-defined characteristics, such as a microspore size below 2 nm, high surface area, well-ordered structure, and negative charge framework. Zeolites have also been used in ion exchange, which might be of the best interest in this study [34,35,36,37]. As such, the zeolite electro-affinity is anticipated to enable the diffusion mechanism and water transportation during the separation process [38,39,40].

Zeolite materials pose a significant hydrophilic character, which is great for composite membrane development [41,42]. Probably, the resulting composite membranes can exhibit excellent performance. Several studies have shown that the development of typical composite membranes with antifouling behavior has demonstrated superior performances in diverse applications such as solvent extraction, pervaporation, desalination, volatile organic extraction from the air, microextraction, the concentration of pharmaceutical waste, and dehumidification [43,44,45,46,47]. 

Polymer membranes can be prepared through a simple method such as phase inversion. In this process, a casting polymer solution is poured on a flat glass surface and cast to form a flat membrane to a required thickness. This is followed by a controlled process of exchanging solvent for a nonsolvent in a water bath where a uniform membrane skin layer is formed [48,49,50]. With this process, typical microfiltration (MF) and ultrafiltration (UF) membranes are mostly produced. The process of using a MF and UF membrane is considered a low-pressure process due to the relatively low operating pressures between the 0.1 to 1.0 MPa range [51,52,53]. The formation of thick and dense membranes is attributed to high separation and selectivity towards macromolecular impurities, such as colloids, emulsions, proteins, bacteria, and viruses [54,55]. Meanwhile, modifications of UF membranes can be effective in the removal of micropollutants without using high-pressure systems.

Understanding the influence of developing selective membranes for utmost performance and being highly selective for salt rejection is highly desirable. As such, this study presents the ZSM-22/PES membranes prepared using zeolite as nanoadditives. The ZSM-22 zeolite presents a well-defined structure with properties of having high selectivity, acidity, negatively charged framework, and hydrothermal stability, which are applicable in important research and reactions in the areas of water treatment, membrane development, petroleum refining, and petrochemicals. Given this, the presented work has differently loaded ZSM-22 nanoadditives at nominal wt.% loadings of 0.1, 0.3, 0.5, 0.75, and 1.0, respectively, from C60-ZSM-22 and F60-ZSM-22 forms. The resulting membranes were prepared via the phase inversion method with the fabrication of the UF-PES membrane into ZSM-22/PES NF membranes as selective saltwater filters. The obtained membrane materials were characterised with XRD, FTIR spectroscopy, and SEM.

## 2. Experimental

### 2.1. Materials

The resulting chemicals and reagents such as colloidal silica, fumed silica, aluminium sulfate hydrate ((Al_2_(SO_4_)_3_.18H_2_O), 1,6-Hexanediamine (HMDA), sodium hydroxide (NaOH), potassium hydroxide (KOH), nitric acid (HNO_3_), aluminium chloride (AlCl_3_), magnesium chloride (MgCl_2_), magnesium sulphate (MgSO_4_), sodium chloride (NaCl), bovine serum albumin (BSA), polyethersulfone (PES) of a 3 mm nominal granule size, and n-methylpyrrolidone (NMP) were acquired from Sigma Aldrich (Johannesburg, RSA) and used as they are.

### 2.2. Methods

The synthesis of ZSM-22 zeolite materials and the preparation of ZSM-22/PES membrane materials followed similar procedures reported in our previous study, except that here, colloidal silica and fumed silica were used as silica sources [23]. The resulting mixture was allowed to stir until a homogenous gel-like product of molar composition, 60SiO_2_:Al_2_O_3_:9KOH:27DAH:3600H_2_O, was attained. This homogeneous gel was then transferred into a bench-top stainless-steel autoclave (300 mL). The autoclave was then allowed to heat to about 160 °C for 6 days for the zeolite crystallisation to occur. After the autoclave was quenched in a cold-water bath, the white solid crystals were obtained by filtration and subsequently washed using deionised water until a neutral pH was reached. The white solid samples obtained were then air-dried for 3 days and removed from the HDMA template via calcination at 550 °C for 24 h. The obtained zeolite materials were denoted as C60-ZSM-22, which refers to colloidal-silica-synthesized ZSM-22 zeolite with Si/Al 60, and F60-ZSM-22, referring to fumed-silica-synthesized ZSM-22 zeolite with Si/Al 60. 

The ZSM-22/PES membranes were modified by the dispersion of the hydrothermally synthesised ZSM-22 powder as nanoadditives into the PES polymer matrix and prepared via a phase inversion method. The PES pellets were allowed to dry at 80 °C for 24 h in the oven. Following, about 18 g of PES pellets were dissolved by 82 mL of an NMP solvent and allowed to stir for 3 h. This was followed by the addition of ZSM-22 nanoadditives at nominal weight loadings of 0, 0.1, 0.3, 0.5, 0.75 and 1.0 wt.% while stirring, respectively, and they were left to stir for a further 24 h. Afterward, the obtained clear homogenous mixture as the membrane casting solution was degassed for about 24 h under a vacuum to allow the dissipation of the trapped air. Prior to casting the membranes, the casting solution was ultra-sonicated for about 3 h to prevent nanoparticle agglomeration in the solution mixture. Then, the casting solution was poured on a clean glass plate and a 200 μm air gap casting knife was used to cast the membranes. The glass plate was let to stand in the air for the 30 s before immersion into the coagulation bath containing deionised water for at least 15 min. This was followed by transferring obtained membranes into another clean deionised water bath to let the membranes cure for 24 h. Thereafter, the membranes were wet with deionised water and stored in plastic bags and kept below 5 °C refrigeration for further analysis and assessment.

### 2.3. Characterization

The X-ray diffraction (XRD) patterns of the ZSM-22 zeolite additives and ZSM-22/PES samples were analysed at a scan step size of 0.025° using a diffractometer PW 3050/60 (XPert-Pro, Almelo, the Netherlands) PANalytical with a PSD Vantec-1 detector and Cu Kα radiation (λ = 1.5406) at a scan speed of degree/second time/step. The attenuated total reflectance Fourier-transform infrared (ATR-FTIR) spectra of the obtained sample materials in this study were generated using a Perkin Elmer Spectrum 100 FTIR spectrometer (Waltham, MA, USA) at a scan range of 400–4000 cm^−1^, at a resolution of 4 cm^−1^, and over an average of 16 scans. The analysis was done using a Bruker Vector 22 mid-IR spectroscopy (Bruker, Karlsruhe, Germany). The morphological features (surface and cross-sectional analysis) and elemental analysis of both zeolite and membrane materials were scanned using the lowest beam current of the scanning electron microscopy-energy dispersive X-ray (SEM/EDX), (Brno-Kohoutovice, Czech Republic) at 5 kV. The optimum resolutions were achieved at a specific magnification. A small amount of zeolite powder on a piece of a membrane was mounted on a sample holder using a carbon tap and was carbon-coated before analysis. In evaluating the membrane surface hydrophilicity or hydrophobicity, a contact angle goniometer (G10, KRUSS, Hamburg, Germany) was used. The water contact angle (WCA) measurements in (°) of the prepared ZSM-22/PES membranes were evaluated at a constant room temperature and 50% moisture utilising a sessile drop method.

### 2.4. Performance Evaluation

A 1000 ppm simulated stock solution of NaCl, MgCl_2_, AlCl_3_, MgSO_4_, and BSA (ca. 1.00 g each) was prepared by dissolving salts/reagents with deionised water in a 1 L volumetric flask. The flasks were then capped and stored below the 5 °C refrigerator (to avoid decomposition of the solution) prior to analysis. The performance indicators of membranes, i.e., pure water flux (using deionised water), salt rejection (using about 1000 ppm of NaCl, MgCl_2_, AlCl_3_, and MgSO_4_), and antifouling (using 1000 ppm protein BSA) were done using an N_2_ gas pressurized Sterlitech (Auburn, AL, USA) dead-end stirred cell with an effective surface area of 1.26 cm^2^. The membrane permeates flux was defined by the following equation:(1)$$P_{\mathrm{wf}}=\frac{Q}{A\times t}$$
where Q is the volume (L) of the pure water flux permeate, while A is the membrane effective surface area (m^2^) of the membrane, and t is the collection time (h) taken for the permeate. The resulting salt rejection was measured using a conductivity meter and calculated using the following equation:(2)$$R\left(\%\right)=\left(1-\frac{C_{p}}{C_{f}}\right)\times 100$$
where C_p_ and C_f_ refer to the concentration of the permeate and the feed, respectively. The antifouling parameters of the membranes were calculated using the following equations:(3)$$\mathrm{Total}\;\mathrm{Fouling}\;\left(\%\right)=\frac{J_{0}-J_{1}}{J_{0}}$$
(4)$$\mathrm{Flux}\;\mathrm{Recovery}\;\mathrm{Ratio}\;\left(\%\right)=\frac{J_{2}}{J_{0}}$$
(5)$$\mathrm{Reversible}\;\mathrm{Fouling}\;\mathrm{Ratio}\;\left(\%\right)=\frac{J_{2}-J_{1}}{J_{0}}$$
(6)$$\mathrm{Irreversible}\;\mathrm{Fouling}\;\mathrm{Ratio}\;\left(\%\right)=\frac{J_{0}-J_{2}}{J_{0}}$$

All the water flux measurements and analyses were done in the study for the resulting membrane materials were at room temperature. Initially, each membrane was compacted until a steady state was reached under the condition of pre-compacting pressure for about 1 h at a pressure of 1.2 bar. Upon attaining a steady flow, the pressure was then lowered to a 1.0 bar constant operating pressure while measuring the permeate water flux (P_WF_) at different time intervals. A similar procedure was repeated with the protein BSA solution. Upon protein fouling, the membrane was turned upside down and backwashed with deionised water for about 1 h. Then, after backwashing, the membrane was reinserted and the pure water flux analysis was repeated using a similar procedure. The salt rejection evaluation also followed a similar procedure of pure water flux analysis and the conductivity of the solutes was measured using a conductivity meter. 

## 3. Results

### 3.1. X-ray Diffraction (XRD) Analysis

The X-ray diffraction patterns of the ZSM-22/PES NF membrane samples prepared via phase inversion at different nominal wt.% loadings are presented in this study. Figure 1 shows the XRD patterns of (a) C60-ZSM-22/PES and (b) F60-ZSM-22/PES membranes with inserted XRD patterns of C60-ZSM-22 and F60-ZSM-22 zeolites as additives, respectively. The inserted C60-ZSM-22 and F60-ZSM-22 XRD patterns, as seen in Figure 1a,b, resemble a typical TON structure as reported in various studies [28,35,56,57]. This is followed by the XRD patterns of PES (shown in Figure 1a or Figure 1b), which exhibit a typical amorphous PES phase. The resulting membranes (C60-ZSM-22/PES and F60-ZSM-22/PES) further exhibit PES amorphousness, which decreases as the zeolite wt.% loading increases. Upon zeolite additives added to the PES matrix, traces of ZSM-22 diffraction peaks can be observed, which correspond with the inserted XRD patterns of C60-ZSM-22 and F60-ZSM-22 zeolites, respectively, in line with other studies [58,59,60]. This confirms the successful incorporation of C60-ZSM-22/F60-ZSM-22 additives into the PES matrix. 

At a 2θ value of around 8°, a 110 diffraction plane attributed to the ZSM-22 TON zeolite can be seen in both Figure 1a,b. However, the F60-ZSM-22/PES NF membranes exhibit a clearer diffraction plane than the counterpart C60-ZSM-22/PES NF membranes as seen. The appearance of zeolite additives onto zeolite membranes has also been observed elsewhere [61,62,63,64]. An improvement in the visibility of the diffraction can also be observed as the wt.% loading increases. Here, as the wt.% loading increases, the peak intensities of the dispersed zeolites into the PES also increased, which confirms the existence of zeolite within the material composition. Further, the diffraction peaks can be observed at 2θ values around 25° for membranes with higher zeolite wt.% loadings. It is conclusive to state that the inclusion of C60-ZSM-22 and F60-ZSM-22 as additives into the PES matrix has been successful. Therefore, F60-ZSM-22/PES exhibits higher prominent existence of both ZSM-22 zeolite and PES than the use of C60-ZSM-22 as additives, which could result due to the differently used silica precursors.

### 3.2. The Attenuated Total Reflectance Fourier-Transform Infrared (ATR-FTIR) Spectroscopy Analysis

Figure 2a,b, respectively, present the ATR-FTIR spectra of the C60-ZSM-22/PES and F60-ZSM-22/PES NF membranes. The resulting ATR-FTIR spectrum reveals typical PES features with intense vibration peaks at 3073 cm^−1^ that can be attributed to the aromatic CH-vibration of a typical pure UF-PES membrane. Upon C60-ZSM-22 and F60-ZSM-22 zeolite additives inclusion in the PES matrix, respectively, the PES features could still be observed. Weaker development of vibrational bands upon C60-ZSM-22 or F60-ZSM-22 addition could suggest that the ZSM-22 additives are buried within the PES polymer matrix. 

The ATR-FITR spectra of either C60-ZSM-22/PES or F60-ZSM-22/PES NF membranes exhibit identical features as that of the PES membrane. Although new shoulder peaks can be observed at 1661 cm^−1^ followed by an intense peak development at 1914 cm^−1^ for both C60 and F60 materials. This could be attributed to the coexistence of C60-ZSM-22 or F60-ZSM-22 in the PES matrix. As such, there exists a successful incorporation of the C60-ZSM-22 and F60-ZSM-22, which is similar to the observation in Figure 1.

### 3.3. Analysis of Membranes Morphology

#### 3.3.1. Scanning Electron Microscopy (SEM) Surface Analysis

The effects of the number of zeolite additives on the morphology of the prepared ZSM-22/PES NF membranes are presented in this study. Figure 3 below displays SEM micrographs of C60-ZSM-22/PES NF membranes. As revealed in Figure 3, the micrographs exhibit a macroporous surface for all membranes, which decreases with increasing C60-ZSM-22 wt.% loadings. 

The pure PES membrane shows large pores on the surface. As seen in the Figure, the SEM micrographs indicate that the addition of C60-ZSM-22 zeolite additives has pretentious significance on the morphological and structural forms of the PES membranes. This can be ascribed to the observed reduction in pore size (as manifested by the formation of smaller pores than in the pure PES substrate) and intensification of the pore densities as wt.% zeolite loadings increase, which is also observed elsewhere [65]. This shows that C60-ZSM-22 zeolite loading into the PES matrix has contributed to the development of membranes with high porosity, which protracts all through the membrane surface. As such, the increase in the zeolite concentration provides more macropores on the surface of the membranes. Meanwhile, on the other hand, higher wt.% zeolite loading seemly causes agglomeration of the zeolite structures on the membrane surface. The agglomerates of C60-ZSM-22 are more visible as the wt.% loading increases, which could impact the possible filtration process. The observed zeolite agglomeration could also suggest that the C60-ZSM-22 zeolite additives can be near the surface during coagulation and remained submerged within the polymer matrix. These observations can be elucidated by considering the NIPS thermodynamics and kinetics aspects [66,67]. Upon increasing the zeolite additives amount from 0.1–1.0 wt.%, the viscosity of casting solutions was increased and thus, the exchange rate diffusion between the NMP as a solvent and aqueous solution as a nonsolvent was suppressed at the highest viscosity of the casting solution [68,69]. Conferring to the kinetic aspect, late demixing results in a membrane with less porosity, few finger-like pores, and a thicker skin layer [70,71,72]. This does not attribute to the observed near-surface zeolite appearance, which is also attributed to higher porosity. This could be due to the hydrophilic nature of zeolite, which facilitates zeolite agglomerates near the surface during membrane formation and the solvent exchange process [73,74]. 

The following Figure 4 shows the SEM micrographs of F60-ZSM-22/PES NF membranes with 0.1, 0.3, 0.5, 0.75, and 1.0 wt.% F60-ZSM-22 loading. The surface structures of the F60-ZSM-22/PES NF membranes are quite porous with no holes or defects, representing a similar structure as shown in Figure 3. A decrease in pore size upon the incorporation of the F60-ZSM-22 zeolite additives into the PES matrix can be attributed to the s similar trend observed in Figure 3.

This shows that the zeolite additives used here were compatible with the PES polymer matrix. As shown in the figure, lower zeolite loading exhibits larger micropores than the higher wt.% loading. This is attributed to a change in pore sizes as the wt.% of zeolite loading increases. The micrographs also reveal a good dispersion of the additives into the polymer film and as the wt.% loading increase, the near-surface zeolite appearance could be observed in agreement with the previously reported works [48,75,76]. Also, the obtained macrostructures exhibit a uniform macroporous distribution on the surface of the obtained membranes. For the 0.5 wt.% F60-ZSM-22 loading, the zeolite additives were appropriately dispersed in the PES polymer matrix and no agglomeration of particles can be observed. 

In general, the zeolite agglomeration can also be more slightly observed in Figure 4 than in Figure 3 as wt.% zeolite loading increases. This agglomeration and deposition of nanoadditives near the top surface of the membranes can relate to differences in the density and polymer of the agglomerates, which became more visible at higher nanoadditive loading. At 1 wt.% F60-ZSM-22/PES zeolite loading, the particles were well-separated and distributed in the membrane. Therefore, it is conclusive to state that regardless of the nature of the silica precursor used during ZSM-22 synthesis, the resulting C60-ZSM-22/PES and F60-ZSM-22/PES NF membranes resemble similar characteristics, which could be attributed to the zeolite features in membranes [77,78].

#### 3.3.2. Scanning Electron Microscopy (SEM) Cross-Sectional Analysis

The cross-sectional micrographs of the C60-ZSM-22/PES NF membranes prepared using C60-ZSM-22 zeolite additives are shown in Figure 5. 

As shown in the figure, the SEM cross-sectional micrographs of the pure and C60-ZSM-22/PES NF membranes exhibit finger-like macrovoids. It is observed that the addition of ZSM-22 zeolite additives in the PES casting solutions changed the macrovoids topology of the resulting membranes. This is well-revealed by the pure PES membrane having wider macrovoids, which are narrowed with increasing wt.% zeolite loadings in agreement with the observation in Figure 3 and Figure 4. 

The cross-sectional micrographs of the F60-ZSM-22/PES NF membranes prepared using F60-ZSM-22 zeolite additives are shown in Figure 6.

As shown in Figure 6, the cross-sectional micrographs reveal different morphologies upon zeolite’s addition. The resulting material exhibits very narrow macrovoids, which become smaller as the zeolite additives loading is increased. These observations can be attributed to the casting solution viscosity, which increases as the filler additive loadings are increased [79]. As such, the zeolite particles are trapped in the skin layer by the increased viscosity, which also prevents the PES chains from shrinkage during the phase inversion process. Accordingly, the polymer chains and ZSM-22 zeolite additives interfacial stresses are then intensified. In a small period upon relaxation of interfacial stress, the pores were formed probably due to the polymer chains shrinkage as compared to ZSM-22 zeolite additives. 

In general, the zeolite additives are visible on the surface (see Figure 3) of the membrane and exhibit uniform dispersion within the membrane matrix, and no sign of agglomeration was spotted on the cross-sectional side. At higher zeolite loadings of zeolite additives (>0.5 wt.%), the ZSM-22 additives form agglomerates probably due to van der Waals forces of particle interaction [80,81,82]. These occurrences could lead to higher porosity and large pore membranes at their surface. However, the obtained membrane exhibits high pore channels with reduced voids as the zeolite content increases, which is consistent with the micropore size shown by the surface micrographs in Figure 5. As such, the agglomeration of the C60-ZSM-22 zeolite additives aggregate did not contribute to the widening of the pore structures. Further analysis of the morphological aspect of the prepared membranes using F60-ZSM-22 zeolite additives was done in comparison with the effect of zeolite on the formation of zeolite-polymer nanofiltration.

#### 3.3.3. Elemental Analysis

The degree of the ZSM-22 zeolite additives’ dispersion onto the membrane matrix was studied through a EDX elemental analysis technique as shown in Figure 7. A selective number of membranes having 0.3 and 1.0 wt.% loadings of ZSM-22 were analysed. As revealed in the Figure, the unmagnified points represent the Si element in the NF membranes. These results indicate that ZSM-22 nanoadditives were evenly dispersed onto the PES polymer matrix of the membrane. 

Besides, the EDX spectrum of the NF membrane is shown in the resulting Figure. EDX spectrum was applied to the top, middle, and bottom sections of the membrane and the quantitative data obtained for each section are in good agreement with the above findings. Significant incorporation of the ZSM-22 could be observed at a higher wt.% loading of 1. The EDX mapping also exhibits a slight relocation of the zeolite additives toward the top layer of the NF membrane during the NIPS before the solidification of the polymer film as shown in Figure 3 and Figure 4. This is also in agreement with the observed XRD, FTIR, and SEM results, which conclude successful incorporation of ZSM-22 and exhibit the significance of using F60-ZSM-22 over C60-ZSM-22 as a potent filler or zeolite additive in polymer membranes. 

### 3.4. Hydrophobicity and Hydrophilicity Analysis

Contact angle measurement is the key indicator for membrane hydrophobicity or hydrophilicity features [83]. Here, the hydrophobicity of the membranes is determined by a higher contact angle value, while a lower contact angle value is associated with higher hydrophilicity of the membrane [84]. As such, membranes retaining higher surface hydrophilicity will generally have a greater ability to entice water molecules and decrease the adsorption of contaminants, which would play a positive role in improving water flux and antifouling ability.

The contact angle measurements of ZSM-22/PES NF membranes along pure PES membranes are presented in Figure 8. As shown in the Figure, the pure PES membrane attained a water contact angle of 77° because of the intrinsic hydrophobic associated with the PES membrane. Upon C60-ZSM-22 and F60-ZSM-22 zeolite additives added to the PES matrix, contact angle measurements of the resulting NF membranes were reduced.

This could be attributed to hydrophilicity attributed to marginal improvement upon zeolite addition. It can be seen that the contact angle measurements decreased from 77° to 70° for the C60-ZSM-22 zeolite and from 77° to 73° for the F60-ZSM-22 zeolite as wt.% loading, respectively, which attribute to about a 7-degree difference. These results indicate that the addition of zeolite additives into the PES matrix has enhanced the hydrophilic features of the PES membrane and was successful in agreement with the aforementioned XRD, FTIR, and SEM analysis. The hydrophilic zeolite additives included in the casting solutions instinctively move to the top surface of the membrane during the phase inversion process to decrease the interface energy while slowly increasing the surface hydrophilicity [85,86,87]. It can be seen that both materials (PES and Zeolite) pose a hydrophilic character, which led to greater interaction during coagulation. Further, the zeolite additive hydrophilicity did not only affect the contact angle measurement but also influenced the pore development as revealed by the change in the pore size and the shape as observed from SEM analysis (see Figure 3 and Figure 4) that resulted in the newly and densely formed macropore channels for water molecule pathways [88,89,90]. Also, the water contact measurement of the C60-ZSM-22/PES NF membranes is lower (high hydrophilicity) than that of the F60-ZSM-22/PES NF membranes due to the nature of the fumed silica used providing alternate pathways for water molecules to penetrate through the membrane.

### 3.5. Membrane Performance Evaluation

#### 3.5.1. Flux Analysis

The performance of the prepared ZSM-22/PES NF membranes using C60-ZSM-22 and F60-ZSM-22 as zeolite additives was evaluated through saltwater filtration studies using sodium chloride (NaCl), magnesium chloride (MgCl_2_), aluminium chloride (AlCl_3_), and magnesium sulphate (MgSO_4_) as modeled saltwater solutions for the salt-rejection studies. Results of the membrane performance and salt rejection studies are presented.

Figure 9 shows the pure water flux of the prepared NF membranes with C60-ZSM-22 and F60-ZSM-22 zeolites at different wt.% loadings, respectively. It can be observed that the addition of zeolite additives into the PES matrix has induced a much higher water flux in comparison with that of the pure PES membrane labelled as either 0 wt.% ZSM-22/PES for both colloidal and fumed silica zeolites (for comparison with the resulting counterparts). This was also rationalized by the hydrophilic features of the zeolite ZSM-22 that provide new flow paths and increase pore density, i.e., decreased tortuosity, for water molecules to penetrate through the composite membrane due to its provided macrospores and diffusion ability. It can be seen that the water flux increased with the filler zeolite loading in the range of 0.1 wt.% and 1.0 wt.% while exhibiting a steady decrease as the time increased. This is similar to the work reported elsewhere [43,91,92,93]. As such, the use of C60-ZSM-22 zeolite additives exhibits decreased water flux more than that of F60-ZSM-22 zeolite additives as contact time is increased. 

The C60-ZSM-22/PES NF membranes demonstrated higher contact angle measurements, which can be attributed to lower hydrophilicity than the F60-ZSM-22/PES NF membranes. This is in agreement with the attained lower water flux as shown in the figure. Meanwhile, F60-ZSM-22/PES membranes demonstrated higher water flux with the observed lower contact angle measurements exhibiting improved hydrophilicity [17,18,19,94]. This lower enhancement of water flux for F60-ZSM-22/PES than C60-ZSM-22 NF by about 150 L·m^−2^·h^−1^ can be explained by the difference in diffusion affinity of the resulting membranes and the inherent hydrophilic [88,89,95,96]. It can be stated that the nature of the silica precursor influences membrane performance. The use of fumed silica seems to have promoted additional flow paths through its hydrophilicity additive, which is not easily understood at this state. The SEM analysis has revealed denser macroporosity for membranes prepared using fumed silica zeolite than those of colloidal silica, which ultimately resulted in the improved water flux. 

The enhanced surface hydrophilicity of the fumed silica zeolite NF membranes attract more water molecules close to the membrane than those of colloidal silica zeolite NF membranes, and facilitated the transport of water molecules in the membrane, which is an observed phenomenon here in line with most reported work [51,97,98]. Also, the diffusion affinity seems to be of great interest probably due to the nature of both zeolites resulting from different silica sources used. It is interesting to note that regardless of the nature of the zeolite used as zeolite additives, the resulting membranes have achieved greater water flux above 500 and 660 L·m^−2^·h^−1^ for both C60-ZSM-22/PES and F60-ZSM-22/PES, respectively.

The observed lower water flux for C60-ZSM-22/PES membranes can be assorted with a poorer affinity of colloidal silica than that of fumed silica [99,100] in agreement with the observed C60-ZSM-22 XRD pattern. It is also clear that the change of water flux is inversely proportional to the trend of the hydrophilic property of the membranes (see Figure 8), indicative of the nanoadditive effects. Membranes with higher hydrophilicity exhibit higher water flux as shown in both Figure 9a,b. As such, the effects of the zeolite additives can also be observed upon increasing zeolite wt.% loading.

#### 3.5.2. Membranes Fouling Analysis

##### Antifouling Analysis

The most known polymer with high thermo-resistance for membrane preparation is polyethersulfone (PES). However, this polymer suffers a major disadvantage concerning its hydrophobic character. Hydrophobicity in the membrane facilitates high adsorption of organic compounds on the membrane surface, which leads to membrane fouling [101,102]. The antifouling performance of C60-ZSM-22/PES and F60-ZSM-22/PES NF membranes is shown in Figure 10.

Membrane fouling can be expressed as the formation of a cake-layer structure, concentration, polarization, and adsorption of the solute on the top surface and pore walls of the membrane [103,104]. As shown in Figure 10, there is a decrease in membrane permeability upon replacing the pure water (deionised water) with the fouling agent BSA. This observation suggests that membrane fouling has occurred in agreement with the reported work [51,105]. The results disclose that the pure 0 wt.% ZSM-22/PES membrane exhibited a high value of water flux reduction to below 200 L·m^−2^·h^−1^ probably due to its high hydrophobic affinity. The 0.3 wt.% loaded membranes also exhibit a low flux of +/−180 L·m^−2^·h^−1^ upon using BSA as a foulant. This is more improved when F60-ZSM-22/PES membranes are used. The most interesting is that the resulting membranes exhibit better water flux after backwashing when compared with the pure PES (0 C60/F60-ZSM-22/PES) membrane. Although in general, it is observed that after backwashing the membranes, the water flux decreased, indicating that there is still a high residual of BSA molecules on the rough surface of the NF membranes, which is a phenomenon observed with most membrane materials upon fouling [11,106,107]. After backwashing of the membranes to remove the attached BSA molecule lower flux increase was observed with increasing wt.% zeolite loading. This is probably due to the improved hydrophilicity of the NF membranes. It is further observed that the NF membranes prepared using fumed silica zeolite still exhibit superior performance to their counterparts of colloidal silica zeolite membranes upon fouling.

##### Flux Recovery and Reversible and Irreversible Fouling

The antifouling characteristics of the C60-ZSM-22/PES and F60-ZSM-22/PES NF membranes were further studied by analysing the membrane flux recovery ratios (FRR), total fouling (R_t_), reversible fouling (R_r_), and irreversible fouling (R_ir_) parameters. The total fouling of the membrane can be expressed as the reduction in water flux because of membrane fouling and higher measurements of total fouling is an indication of severe membrane fouling.

The flux recovery ratios of C60-ZSM-22/PES and F60-ZSM-22/PES NF membranes are shown in Figure 11. It can be observed in both Figure 11a,b that the flux recovery ratios of the resulting membranes increase with increasing zeolite wt.% loading. This can be explained as an improvement of the antifouling character of the NF membranes upon zeolite as nanoadditives included in the PES matrix. The PES membranes exhibit a lower recovery ratio probably due to their inherited poor hydrophilicity, which is associated with typical PES polymers.

Both the flux recovery ratio of C60-ZSM-22/PES and F60-ZSM-22/PES NF exhibit improvement from 42% to about 80%. Here, the C60-ZSM-22/PESS NF exhibits better antifouling than the F60-ZSM-22/PES counterpart. It was found in Figure 10 that when compared with the pure PES membrane, the resulting NF membranes exhibit smaller fouling values. This can be expressed by the total fouling analysis shown in Figure 12.

As shown in Figure 12, the results further indicate that the C60-ZSM-22/PES and F60-ZSM-22/PES NF composite membranes have better antifouling performance than the pure PES membranes. The high total fouling exhibited by the pure PES membrane evidenced this. It is interesting to note that the total fouling exhibited by C60-ZSM-22/PES and F60-ZSM-22/PES decreased with increasing zeolite wt.% loading. Zeolite loadings of 1.0 wt.% for both C60-ZSM-/PES and F60-ZSM-22/PES NF attained lower total fouling of 63 and 64%, respectively. The following reversible fouling graphs for C60-ZSM-22/PES and F60-ZSM-22/PES are shown in Figure 13. 

As usual, the pure PES membranes exhibit lower reversible fouling. However, the resulting NF membranes as well exhibit lower reversibility as the zeolite wt.% loading is increased. Reversible BSA protein molecule adsorption causes reversible fouling that can be removed by hydraulic cleaning through membrane backwashing. The obtained NF membranes exhibit higher reversibility at lower zeolite wt.% loadings, which is an indication that the obtained membrane can be reused upon backwashing. Figure 14 shows the irreversible fouling graphs for both C60-ZSM-22/PES and F60-ZSM-22/PES NF membranes. 

The lower ratio of irreversible fouling to total fouling of a membrane indicates that fouling can be physically removed through a cleaning process such as backwashing. However, when membrane fouling can only be overcome by the use of chemical reagents, it is therefore defined as irreversible fouling as shown in Figure 14. Here, the 0.3 wt.% NF membranes exhibit a lower ratio of irrepressible fouling, which is in agreement with observed flux recovery ratios (see Figure 11) and total fouling (see Figure 12 and reversible fouling) (see Figure 13). It can be noted that at 0.3 wt.% loadings, the NF exhibits minimum performance as shown by the good hydrophilicity attainment (see Figure 8) and pure water fluxes (see Figure 9). It can be conclusive to state that the 0.3 wt.% C60-ZSM-22/PES and F60-ZSM-22/PES NF membranes are prominent membranes for the desalination of brackish and seawater.

#### 3.5.3. Saltwater Rejection Analysis

In this study, modeled saltwater for brackish and seawater was assessed using water solutions of NaCl, MgCl_2_, AlCl_3_, and MgSO_4_ prepared in our laboratory for membrane performance evaluation. Figure 15, shows saltwater rejection by C60-ZSM-22/PES and F60-ZSM-22/PES NF membranes.

As shown in Figure 15, the pure PES membranes exhibit a salt rejection below 40% for all assessed salts, which amounts to larger macrovoids and poor hydrophilicity for effective water molecule attraction. The C60-ZSM-22/PES NF membranes exhibit higher salts rejection for monovalent salt, which decreases as the salts state increases. Even when the zeolite wt.% loading increases, salt rejection seemly decreases upon changing the state of the salt in the order: NaCl > MgCl_2_ >AlCl_3_ > MgSO_4_. (as manifested by Figure 15a). However, the resulting NF membranes attained above 50% salt rejection, which increases with increasing zeolite wt.% loadings to about 65%. Meanwhile, the salt rejection by all F60-ZSM-22/PES NF membranes was also effective with salt rejection consistently above 50 to 70%. Concurrently, with increasing salt states (i.e., from monovalent to trivalent), the salt rejection was also observed to increase with increasing zeolite wt.%. The salt rejection was enhanced with the addition of zeolite nanoadditives as compared to the pure PES membrane. The observed performance by F60-ZSM-22/PES NF membranes can be attributed to improved hydrophilicity upon the zeolite addition as shown in Figure 8. This was expected for F60-ZSM-22/PES NF membranes to exhibit higher salt rejection than their counterpart C60-ZSM-22/PES. In Figure 9, the F60-ZSM-22/PES NF membranes have shown a higher water flux attributed to the high affinity associated with the fumed silica zeolite, thus having an effective rejection through this specific interaction. 

## 4. Discussion 

The XRD patterns of ZSM-22/PES NF membranes given in Figure 1 exhibited the coexistence of the typical PES matrix as the principal material. Amorphousness was associated with PES materials as they are not crystalline. However, upon addition of ZSM-22 as additives, different diffraction peaks due to ZSM-22 addition could be observed at higher 2θ, and the crystalline phase due to zeolite inclusion can be observed. No diffraction plane was observed at some 2θ peak position, probably due to the zeolite submerged within the polymer matrix. It can also be deduced based on different XRD patterns in Figure 1a,b that the nature of zeolite resulting from different silica sources used has influenced the emendation in the PES matrix, hence the observed XRD pattern. As seen in Figure 2, the inclusion of the ZSM-22 into the PES matrix could be noticed. This could be evidenced by the widening vibration peaks, which increased with increasing wt.% loadings. After the incorporation of C60-ZSM-22 and F60-ZSM-22 nanoadditives, the intensity and peak position of the PES was not compromised, probably because of the relatively low additive concentration in the precursor solution mixture. In Figure 3 and Figure 4, the micrographs indicated that the presence of ZSM-22 zeolite in the PES polymer matrix reduced the pore size and increased the density of the pores as the wt.% zeolite loadings were increased. It can also be stated that the use of ultrasonic waves has caused a uniform dispersion of nanoadditives, and increased pores content with increasing ZSM-22 content as additives could be observed. However, when ZSM-22 wt.% loading exceeds 0.5, agglomerated particles and particle aggregation became more apparent and could not avoid agglomeration. As such, the effect of zeolite inclusion into a polymer matrix can be noticed and evidenced by its coexistence of agglomerate even upon sonicating. As shown in Figure 7, the resulting NF membranes exhibited improved hydrophilicity, which could be attributed to the incorporated ZSM-22 additives. The pure PES membrane showed poor hydrophilicity due to the inherited hydrophobic nature. As shown in Figure 8, there was a much greater decrease in water flux membranes as the wt.% loading was increased more for C60-ZSM-22/PES NF membranes than that of the F60-ZSM-22/PES counterparts. In this respect, membranes with higher surface hydrophilicity have greater water flux. The attained microstructure with internal and dense micropores formed on ZSM-22/PES NF membranes observed from the cross-section of the membranes (as shown in Figure 5 and Figure 6) can reduce the resistance to water molecules and provide additional space for water transportation. Besides, the macroporous nature of zeolite ZSM-22 seemly offered additional water pathways whilst reducing tortuosity and enhancing permeate flux. The results demonstrated that the ZSM-22 zeolite is a suitable additive material for the enhancement of the UF-PES membrane features. Antifouling behavior as demonstrated in Figure 10 supports the effective use of ZSM-22 as an additive filler for NF membrane preparation. Upon BSA fouling, the resulting ZSM-22/PES NF showed a higher water flux than the pure PES membrane from 150–550 L·m^2^·h^−1^ (for C60-ZSM-22/PES) and 150–600 L·m^2^·h^−1^ (for F60-ZSM-22/PES). In Figure 11, higher flux recovery ratios of about 80% were observed for both C60-ZSM-22/PES and F60-ZSM-22/PES NF membranes. These materials further exhibited lower total fouling of 65% as seen in Figure 12, which could amount to the generated hydrophilicity. Reversible fouling in Figure 13, was observed to decrease with increasing wt.% loading for the C60-ZSM-22/PES NF membrane to about 55–60%. This was also observed for F60-ZSM-22/PES NF membranes except at 0.1 wt.% loadings. A characteristic of lower irreversibility to about 20% upon fouling could be seen in Figure 14. The electrostatic interaction of the salts with the surface of the resulting membranes showed much greater effectiveness on F60-ZSM-22/PES NF membranes. Also, the smaller generated macrovoids upon zeolite addition could be attributed to the observed higher salt rejection. It was shown that the salt rejection increases with increasing zeolite wt.% loading and the nature of salts investigated. Figure 15a showed that C60-ZSM-22/PES gave a salt rejection of NaCl and MgCl_2_ to above 60% from the zeolite wt.% of 0.3 at a 1 bar pressure system. A similar trend was also observed for F60-ZSM-22/PES, which also exhibited salt rejection above 50%. The rejection of MgSO_4_ was at about 65% for F60-ZSM-22/PES, suggesting the effectiveness of the NF membranes for MgSO_4_ rejection. The observed salt rejection results are promising for use in the desalination of brackish and seawater processes. This could be explained based on the steric hindrance mechanism since the prepared ZSM-22/PES NF membranes have relatively small pore sizes, as observed in Figure 3 and Figure 4.

## 5. Conclusions

In summary, the ZSM-22/Polyethersulfone (ZSM-22/PES) nanofiltration membranes, containing 0.1 to 1.0 wt.% of C60-ZSM-22 and F60-ZSM-22 as zeolite additives were prepared successfully. The obtained results revealed the formation of nanofiltration membranes with a porous surface and finger-like macrovoid structures for both types of zeolite. The water contact angle measurements became apparent that the zeolite inclusion in the polymer matrix increased the hydrophilicity and the water permeation of the resulting NF membranes. As the pore sizes were reduced with increasing pore density of the NF membranes in the presence of ZSM-22 addition, this provided an improvement in the water permeates flow of the NF membranes with increasing zeolite wt.% loading and time. The initial permeability of these nanofiltration membranes was observed to improve from about 200 L·m^2^·h^−1^ (PES) to above 700 L·m^−2^·h^−1^ at 1 bar. All nanofiltration membranes exhibited excellent antifouling performances as shown through an excellent flux recovery ratio beyond 75%. Total fouling was observed to decrease with increasing wt.% loading to about 65%. Backwashing of the NF membranes attained about 50% reusability of the membranes and exhibited about 20% irreversibility at higher zeolite wt.% loadings. In return, the nanofiltration membranes also successfully rejected salt solutes solely through a size-exclusion mechanism above 50% to a high salt rejection of 70%. Higher effective MgSO_4_ rejection was also observed for F60-ZSM-22/PES nanofiltration membranes. Therefore, the C60-ZSM-22 and F60-ZSM-22 acted by fabricating both the surface morphology and the hydrophilicity of the ultrafiltration polyethersulfone membrane, improving the permeation flow and salt rejection. 

## Figures and Tables

**Figure 1 membranes-12-00553-f001:**
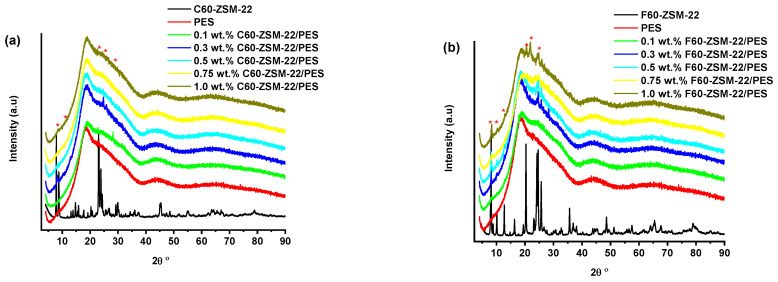
X-ray diffraction patterns of nanofiltration membranes prepared through a phase inversion method to a nominal wt.% loadings of 0, 0.1, 0.3, 0.5, 0.75, and 1.0 using ZSM-22: (**a**) C60-ZSM-22/PES and (**b**) F60-ZSM-22/PES with zeolite as additives.

**Figure 2 membranes-12-00553-f002:**
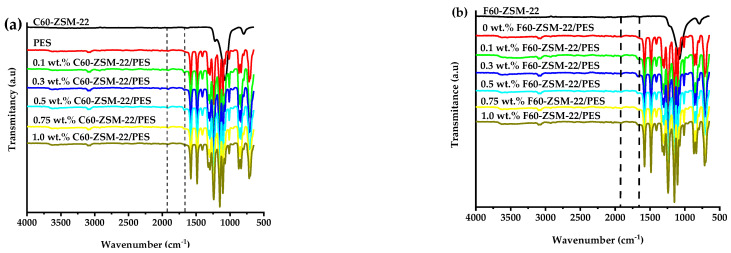
ATR-FTIR spectra of nanofiltration membranes prepared through a phase inversion method to nominal wt.% loadings of 0, 0.1, 0.3, 0.5, 0.75, and 1.0 using ZSM-22: (**a**) C60-ZSM-22/PES and (**b**) F60-ZSM-22/PES with zeolite as additives.

**Figure 3 membranes-12-00553-f003:**
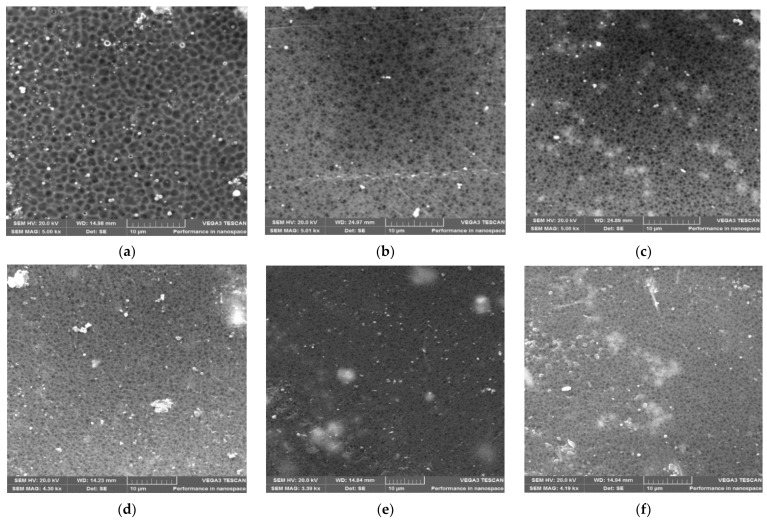
SEM surface micrographs of C60-ZSM-22/PES membrane materials prepared using C60-ZSM-22 as zeolite additives: (**a**) PES, (**b**) 0.1 wt.% C60-ZSM-22/PES, (**c**) 0.3 wt.% C60-ZSM-22/PES, (**d**) 0.5 wt.% C60-ZSM-22/PES, (**e**) 0.75 wt.% C60-ZSM-22/PES, and (**f**) 1.0 wt.% C60-ZSM-22/PES via phase inversion.

**Figure 4 membranes-12-00553-f004:**
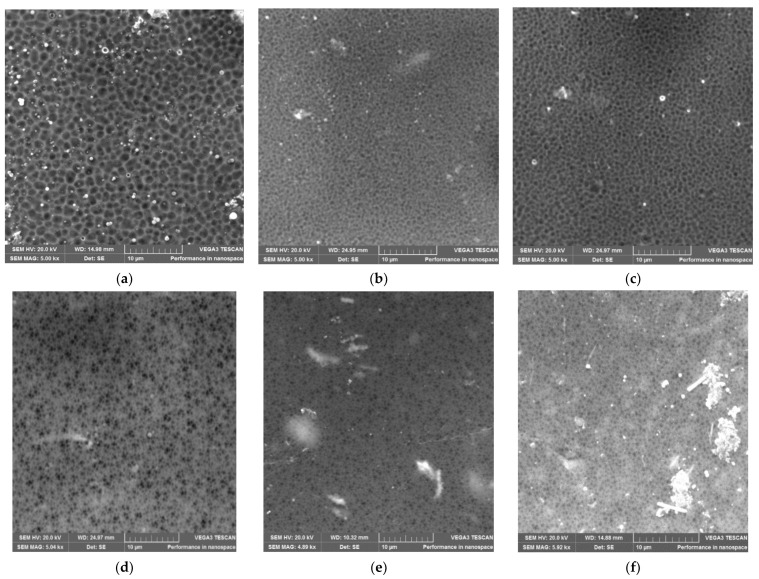
SEM surface micrographs of F60-ZSM-22/PES nanofiltration membrane materials prepared using F60-ZSM-22 as zeolite additives: (**a**) PES, (**b**) 0.1 wt.% F60-ZSM-22/PES, (**c**) 0.3 wt.% F60-ZSM-22/PES, (**d**) 0.5 wt.% F60-ZSM-22/PES, (**e**) 0.75 wt.% F60-ZSM-22/PES, and (**f**) 1.0 wt.% F60-ZSM-22/PES via phase inversion.

**Figure 5 membranes-12-00553-f005:**
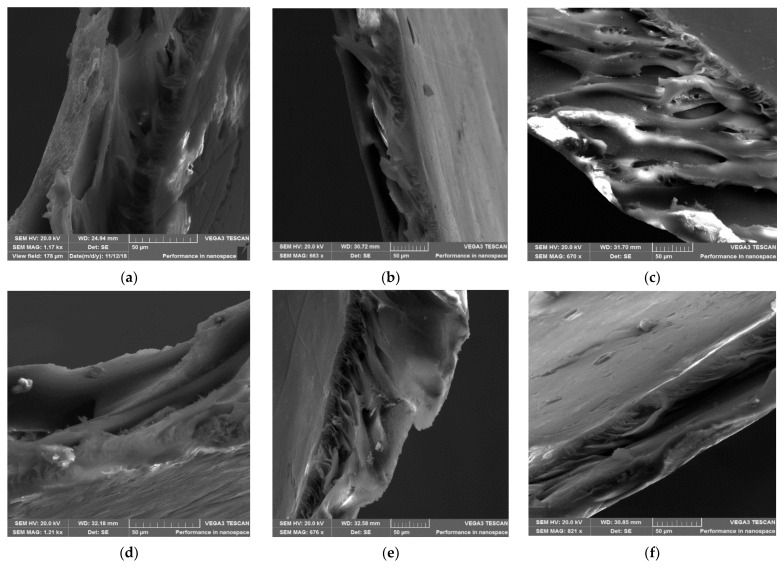
SEM cross-sectional of C60-ZSM-22/PES membrane materials prepared using C60-ZSM-22 as zeolite additives: (**a**) PES, (**b**) 0.1 wt.% C60-ZSM-22/PES, (**c**) 0.3 wt.% C60-ZSM-22/PES, (**d**) 0.5 wt.% C60-ZSM-22/PES, (**e**) 0.75 wt.% C60-ZSM-22/PES, and (**f**) 1.0 wt.% C60-ZSM-22/PES via phase inversion.

**Figure 6 membranes-12-00553-f006:**
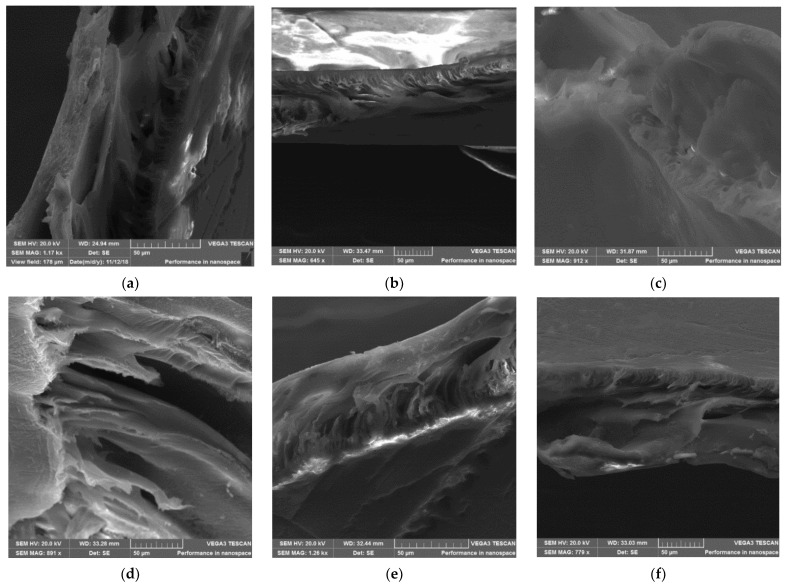
SEM cross-sectional of F60-ZSM-22/PES membrane materials prepared using F60-ZSM-22 as zeolite additives: (**a**) PES, (**b**) 0.1 wt.% F60-ZSM-22/PES, (**c**) 0.3 wt.% F60-ZSM-22/PES, (**d**) 0.5 wt.% F60-ZSM-22/PES, (**e**) 0.75 wt.% F60-ZSM-22/PES, and (**f**) 1.0 wt.% F60-ZSM-22/PES via phase inversion.

**Figure 7 membranes-12-00553-f007:**
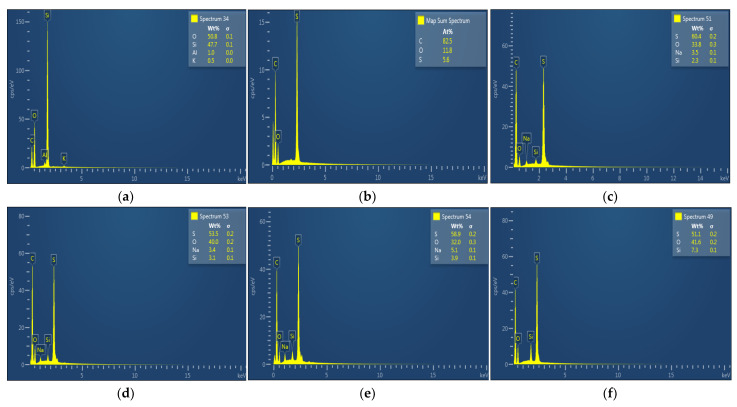
EDX spectra of (**a**) ZSM-22, (**b**) PES, (**c**) 0.3 wt.% C60-ZSM-22/PES, (**d**) 0.3 wt.% F60-ZSM-22/PES, (**e**) 1.0 wt.% C60-ZSM-22/PES, and (**f**) 1.0 wt.% F60-ZSM-22/PES membrane materials.

**Figure 8 membranes-12-00553-f008:**
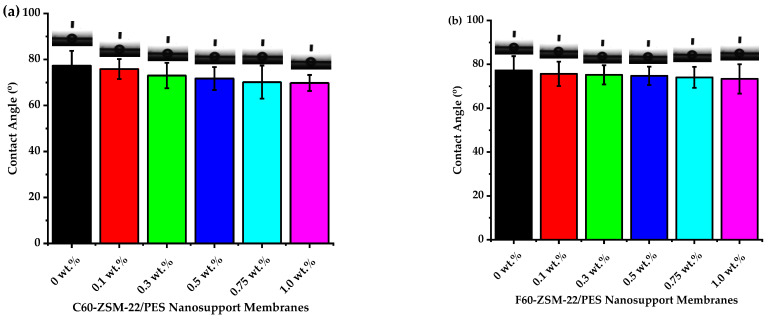
Contact angle measurements of nanofiltration membranes prepared via phase inversion to nominal wt.% loadings of 0, 0.1, 0.3, 0.5, 0.75, and 1.0 using ZSM-22: (**a**) C60-ZSM-22/PES and (**b**) F60-ZSM-22/PES as zeolite additives.

**Figure 9 membranes-12-00553-f009:**
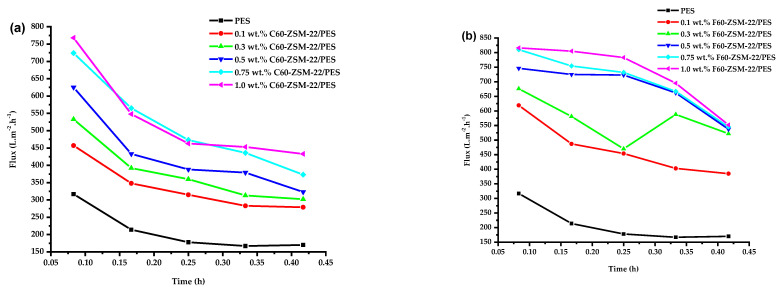
Pure water flux analysis of nanofiltration membranes prepared via phase inversion to nominal wt.% loadings of 0, 0.1, 0.3, 0.5, 0.75, and 1.0 using ZSM-22 zeolite as nanoadditives: (**a**) C60-ZSM-22/PES and (**b**) F60-ZSM-22/PES.

**Figure 10 membranes-12-00553-f010:**
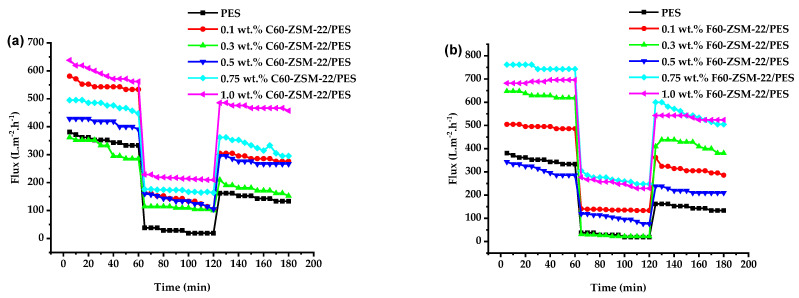
Antifouling analysis of nanofiltration membranes prepared via phase inversion to nominal wt.% loadings of 0, 0.1, 0.3, 0.5, 0.75, and 1.0 using ZSM-22 zeolite as nanoadditives: (**a**) C60-ZSM-22/PES and (**b**) F60-ZSM-22/PES.

**Figure 11 membranes-12-00553-f011:**
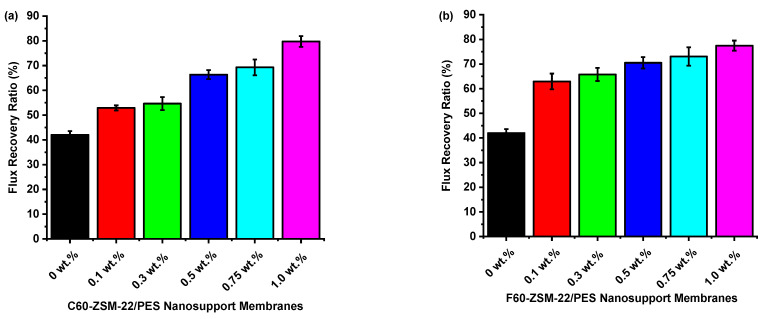
Flux recovery ratios analysis of nanofiltration membranes prepared via phase inversion to nominal wt.% loadings of 0, 0.1, 0.3, 0.5, 0.75, and 1.0 using ZSM-22 zeolite as nanoadditives: (**a**) C60-ZSM-22/PES and (**b**) F60-ZSM-22/PES.

**Figure 12 membranes-12-00553-f012:**
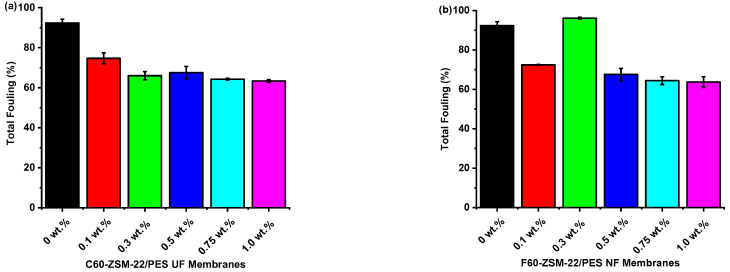
Total fouling analysis of nanofiltration membranes prepared via phase inversion to nominal wt.% loadings of 0, 0.1, 0.3, 0.5, 0.75, and 1.0 using ZSM-22 zeolite as nanoadditives: (**a**) C60-ZSM-22/PES and (**b**) F60-ZSM-22/PES.

**Figure 13 membranes-12-00553-f013:**
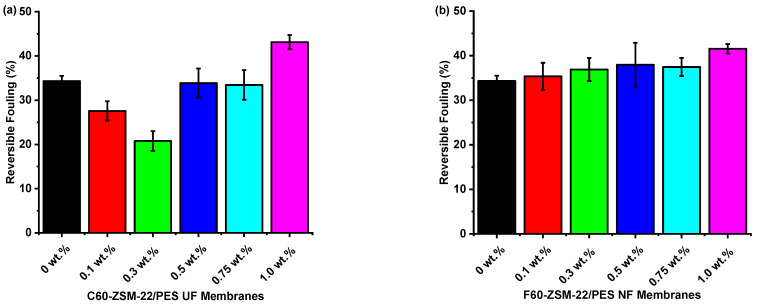
Reversible fouling analysis of nanofiltration membranes prepared via phase inversion to nominal wt.% loadings of 0, 0.1, 0.3, 0.5, 0.75, and 1.0 using ZSM-22 zeolite as nanoadditives: (**a**) C60-ZSM-22/PES and (**b**) F60-ZSM-22/PES.

**Figure 14 membranes-12-00553-f014:**
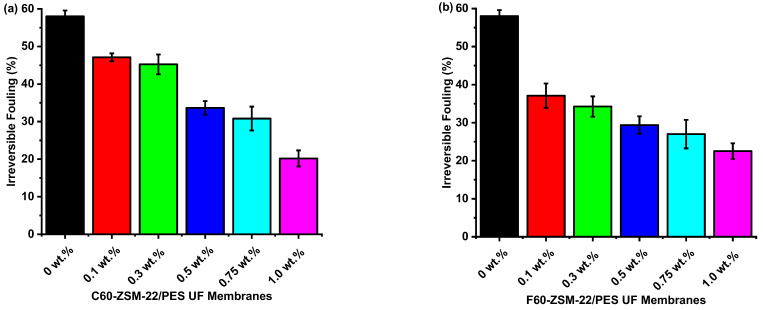
Irreversible fouling analysis of nanofiltration membranes prepared via phase inversion to nominal wt.% loadings of 0, 0.1, 0.3, 0.5, 0.75, and 1.0 using ZSM-22 zeolite as nanoadditives: (**a**) C60-ZSM-22/PES and (**b**) F60-ZSM-22/PES.

**Figure 15 membranes-12-00553-f015:**
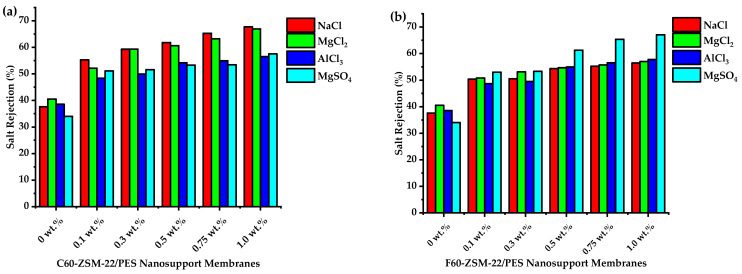
Salt rejection analysis of nanofiltration membranes prepared via phase inversion to nominal wt.% loadings of 0, 0.1, 0.3, 0.5, 0.75, and 1.0 using ZSM-22 zeolite as nanoadditives: (**a**) C60-ZSM-22/PES and (**b**) F60-ZSM-22/PES.

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
