# Peer review of "Influence of the Zeolite ZSM-22 Precursor on a UF-PES Selective Substrate Layer for Salts Rejection"

_membranes, 2022, doi:10.3390/membranes12060553_

Round 1
Reviewer 1 Report
This paper described preparation of ZSM-22/PES membrane and its separation properties. It was found that the addition of ZSM-22 zeolite materials is helpful to improve its separation performance.Since the reported results were meaningful in this field, I recommend it to be published after minor revision.
(1)The abstract of the manuscript is not refined enough to highlight the innovation. It may be better to let the advantages of the concept can be understood clearly.
(2)The scientific discussion and comparison for other research should be added.
(3)The mechanism of the membrane formation is missing from the introduction section.
Author Response
Reviewer 1
This paper described the preparation of the ZSM-22/PES membrane and its separation properties. It was found that the addition of ZSM-22 zeolite materials is helpful to improve its separation performance. Since the reported results were meaningful in this field, I recommend it to be published after minor revision.
Point 1: The abstract of the manuscript is not refined enough to highlight the innovation. It may be better to let the advantages of the concept can be understood clearly.
Response 1: Line 12-32, Rephrased the abstract to “fabrication of the ZSM-22/Polyethersulfone (ZSM-22/PES) membranes as selective salt filters represent a growing membrane technological area in separation with the potential of high economic reward based on its low energy requirements. The incorporation of ZSM-22 zeolite material as additives into the PES polymer matrix has the prospective advantage of examining both the zeolite and polymer features while overcoming the limitations associated with both materials. This work investigates the influence of the nature of the silica precursor on ZSM-22 zeolite hydrothermally synthesised using colloidal (C60) and fumed (C60) silica to Si/Al of 60. Characterisation of these materials with an X-ray Diffraction (XRD), Fourier Transform Infrared (FTIR) and Scanning Electron Microscopy (SEM) with Energy Dispersive X-Ray (EDX) Analysis revealed successful synthesis, where highly crystalline materials were obtained. The ZSM-22 additives were directly dispersed into a PES polymeric matrix to form a casting solution for the preparation of the ZSM-22/PES nanofiltration (NF) selective substrate layers via a phase inversion method for salts rejection. The polymeric PES was selected as an organic network in which the content of the ZSM-22 zeolite (ranging between 0 and 1.0 wt.%), was obtained and characterised by XRD, FTIR, and SEM analysis, as well as water contact angle (WCA) measurement and dead-end filtration cell. The phase inversion preparation method has induced the resulting ZSM-22/PES NF substrates anisotropy as attributed to high water flux to above 700 L.m-2.h-1 and salt rejection beyond 80% as revealed by the obtained results. The materials also exhibited improved antifouling behaviour to above 70% flux recovery ratios. As such, the nature of the silica precursor influences ZSM-22 zeolite as additives in PES polymer matrix and led to the enhanced performance of the PES ultrafiltration membrane”.
Point 2: The scientific discussion and comparison for other research should be added.
Response 2: New references and comparison with other research work was updated in section 3 and 4.
Point 3: The mechanism of the membrane formation is missing from the introduction section.
Response 3: Line 95-106, The following statement was added, “Polymer membranes can be prepared through simple methods such as phase inversion. In this process, a casting polymer solution is poured on a flat glass surface and cast to form a flat membrane to a required thickness. This is followed by a controlled process of exchanging solvent for a nonsolvent in a water bath where a uniform membrane skin layer is formed [43-45]. With this process, typical microfiltration (MF) and ultrafiltration (UF) membranes are mostly produced. The process of using MF and UF membrane is considered a low-pressure process due to their relatively low operating pressures between 0.1 to 1.0 MPa ranges [46-48]. The formation of think and dense membranes is attributed to high separation and selectivity towards macromolecular impurities, such as colloids, emulsions, proteins, bacteria, and viruses [49, 50]. Meanwhile, modifications of UF membranes can be effective in the removal of micropollutants without using high-pressure systems”.

Author Response
Reviewer 2
This paper reports the addition of zeolites from two different precursors to improve the NF permeance of the PES membranes. The manuscript requires further polishing in terms of structure and language. More importantly, provided evidence and explanations are not convincing to support the exceptionally high flux and separation performances claimed by the authors.
Major comments:
Point 1: Cross-sectional images are not clear, especially in Figures 5a, 6c,6d,
Response 1: Line 327-346, Adjusted the resolution of the micrographs, Figure 5(a) is similar to Figure 6(a), however, the cross-sectional images are reported at a similar scale.
The corresponding supplementary image to Figure 6(c) and (d), respectively are included below to expand the observation.
Point 2: Some discussions seem to be repetitive in section 3.3.1, nanoparticle concentration/density vs pore structure explanation, nanoparticle agglomeration at the surface. Authors need to be more precise.
Response 2: Line 260-326, Improved the narratives and eliminated repetitive words.
Point 3: Line 388… “However, the water flux decreased with the filler zeolite loading in the range of 0.1 wt.% and 1.0 wt.%” but the graph shows an increase in flux by zeolite addition.
Response 3: Line 438-442, Rephrased the statement, “It can be seen that the water flux increased with the filler zeolite loading in the range of 0.1 wt.% and 1.0 wt.% while exhibiting a steady decrease as the time increased. This is similar to the work reported elsewhere [43, 90-92]. As such, the use of C60-ZSM-22 zeolite additives exhibits decreased water flux than that of F60-ZSM-22 zeolite additives as contact time is increased”.
Point 4: Why there is a spike in flux for 0.3 wt.% F60-ZSM-22/PES from 0.25h to 0.35h? The experimental section mentioned flux is evaluated at a steady state but the presented flux still varies with time. It is more meaningful to report steady-state flux.
Response 4: Figure 9(b), although the evaluation was conducted under steady pressure the membranes pore structures are not uniform and will respond differently. As such, the spike increase can be explained as due to the slightly observed voids as shown in Figure 4 (c), which is similar to the pure PES membrane.
Point 5: SEM revealed pores ranging in a few hundred nanometers yet maintained NaCl rejection above 30% for membranes. Does this membrane have a macroporous morphology on top and dense inside? Provided discussion and evidence are not sufficient to support this claim.
Response 5: Line 571-598, it is observed that salt rejection is not dependent on the porousness of the substrates only, however, also the nature of the membrane itself has an impact. The use of negatively charged zeolite has been attributed to the electro-affinity of the membrane, which is important during electro-static repulsion.
Point 6: Reported flux for composite membranes was unbelievable highly > 500 LMH tested at 1 bar with rejection for NaCl above 50%. Authors need to explain in detail about this exceptional flux.
Response 6: Line 571-598, the approach was to design membranes that will require less pressure to treat water samples. As such more dense porous membranes were developed to create more pathways for water channels. While also trying to improve their selectivity by compressing/reducing pore sizes. The added negatively-charged zeolites also attribute to high rejection since there also exists electrostatic interaction.
Point 7: Minor Comments: 7. Line 15… ‘combing’?
Response 7: Combing was changed to examining, which refers to the evaluation of the membranes.
Point 8: Line 108.. ‘dissolute’
Response 8: Typing error, dissolute was changed to dissolve.
Point 9: The introduction will look better if they present a brief literature review first and then state why they chose ZSM-22.
Response 9: Line 71-88, Rephrased the following paragraph, “In our previous study, ZSM-22 zeolite synthesised using Tetraethylorthosilicate (TEOS), was used in the development of ZSM-22/Polyethersulfone membranes [23]. It was found that this material exhibited a negatively charged framework, well-defined structure and high crystallinity upon synthesis, which was effective in the development of the resulting NF membranes. However, understanding of the influence of nature of silica precursor has not been detailed explored and established. The nature of the silica precursor seems to have an impact on the crystallinity and retention of the negative charge framework in zeolite [24-27]. As such, this study seeks to explore how the nature of silica precursors in ZSM-22 synthesis will influence the performance of the resulting membrane in salt rejections. The ZSM-22 zeolite synthesis is commonly influenced by the nature of the silica precursor, pH of the solution, temperature and time [28-30]. As such prior optimisation of the zeolite synthesis conditions is paramount [31-33]. Here ZSM-22 as nanoadditive fillers was chosen based on its limited work on membrane fabrication and owing to its well-defined characteristics such as microspore size below 2 nm, high surface area, well-order structure and negatively charged framework. Zeolites have also been used in ion exchange, which might be of the best interest in this study [34-37]. As such, the zeolite electro-affinity is anticipated to enable diffusion mechanism and water transportation during the separation process [38-40]”.
Point 10: Authors need to clarify the difference between their previous work of incorporating the same kind of zeolite in the PES in the introduction itself. It is mentioned in the experimental section that a different precursor is used here. Does it make any difference in the properties of zeolites formed?
Respond 10: The question/comment was clarified in response 9.
Point 11: Line 99… ‘batch-top stainless-steel autoclave’. batch-top or bench-top?
Response 11: Batch-top was corrected to bench-top.
Point 12: Please define C60-ZSM-22 and F60-ZSM-22
Respond 12: C60-ZSM-22 refers to Colloidal Silica synthesized ZSM-22 zeolite with Si/Al 60 and F60-ZSM-22 refers to Fumed Silica synthesized ZSM-22 zeolite with Si/Al 60.
Line 134-137, added the following statement; ” The obtained zeolite materials were denoted as C60-ZSM-22, which refers to Colloidal Silica synthesized ZSM-22 zeolite with Si/Al 60 and F60-ZSM-22 referring to Fumed Silica synthesized ZSM-22 zeolite with Si/Al 60”.
Point 13: Can you explain why there are no obvious peaks for C60-ZSM-22 compared to F60-ZSM-22 in the XRD plots despite both of them being at the same concentration? Some unidentified peaks are also visible at C60-ZSM-22 incorporated membranes at lower concentrations.
Response 13: Figure 1, As shown in Figure 1, F60-ZSM-22 exhibit a well-defined and highly crystalline zeolite material than C60-ZSM-22. This could have been attributed to the observed absence of XRD peaks upon inclusion into the PES solution. SiO2 is inherently hydrophilic. The nature of silicate material hydrophilicity is due to the attachment of hydroxyl groups onto the atoms. This makes it dispersible in water, which was crucial during membrane preparation. With C60-ZSM-22 exhibiting less crystallinity this has limited its hydrophilicity leading to few zeolite additives near the membrane surface. The undefined peaks are due to the nature of the C60-ZSM-22.
Point 14: Line 252, 260,265… says microporous but SEM looks macroporous?
Response 14: Corrected to macroporous.
Point 15: Line 323… magnificent??
Response 15: Typing error, corrected to unmagnified referring to lower peaks.
Point 16: Figure 7 is named from b-g while captioning from a-f
Response 16: Typing error, corrected to a-f.
Point 17: Figure 9a unit of flux is incorrect
Response 17: Typing error, Figure 9 (a) flux unit was corrected.

Reviewer 3 Report
Dear Authors,
I have made the following comments and suggestions to improve the quality of the paper:
From lines 33 to 58: Writing only by words without numbers (facts, data). For example:
Line 38. "NF and reverse osmosis (RO) process have promising and efficient production," What is efficiency? How much energy is needed per m3?
Line 40: "The major limitation in the membrane system, which affects the life performance of the membranes, is surface fouling," How much life performance is decreased due to surface fouling (in percentage, %)?
Line 44: How much reduction in numbers (or in %) for water flux, salt rejection, permeability?
Similarly, other sentences in the introduction need numbers, not just words.
Line 189: Values of the vertical axis are missing;
Line 203: Values of the vertical axis are missing;
Line 401: Units of the vertical axis are incorrect;
Line 427: Units of the vertical axis are wrong;
Line 427: Data in figure 10 need to be checked as it is over against data from figure 9.
Line 433: Within which range the water flux was reduced ( need to show in %).
Line 450: Formulas for flux recovery ratios (FRR), total fouling (Rt), reversible fouling (Rr), and irreversible fouling (Rir) are needed.
Line 608: Units are incorrect.
Thanks,
Regards,
Reviewer
Author Response
Reviewer 3.
I have made the following comments and suggestions to improve the quality of the paper:
From lines 33 to 58: Writing only by words without numbers (facts, data). For example:
Point 1: Line 38. "NF and reverse osmosis (RO) process have promising and efficient production," What is efficiency? How much energy is needed per m3?
Response 1: The efficiency production refers to the capability of the resulting materials to produce clean water. NF and RO membrane process can be effective to above 50% salt rejection. The driving force in membrane filtration or water flux is pressure where a dead-end filtration cell can be used.
Point 2: Line 40: "The major limitation in the membrane system, which affects the life performance of the membranes, is surface fouling," How much life performance is decreased due to surface fouling (in percentage, %)?
Response 2: Membrane fouling is a process by which the particles, colloidal particles, or solute macromolecules are deposited or adsorbed onto the membrane pores or a membrane surface by physical and chemical interactions or mechanical action, which results in smaller or blocked membrane pores affecting membrane performance. The decline in life span performance of the membrane due to fouling depends on the nature of fouling. However, the life performance of the membranes can decrease to less than 10%
Point 3: Line 44: How much reduction in numbers (or in %) for water flux, salt rejection, and permeability?
Similarly, other sentences in the introduction need numbers, not just words.
Response 3: Improved the statement by rephrasing it to, “as reported, saltwater desalination through semipermeable membranes viz., NF and reverse osmosis (RO) process have promising and efficient production (with about >50% removal efficiency) of portable fresh and clean drinking water [7-10]. However, these membranes suffer from some limitations. The major limitation in the membrane system, which affects the life performance of the membranes to lower than 10%, is surface fouling [11-13]”.
Point 4: Line 189: Values of the vertical axis are missing;
Response 4: Here the graphs were superimposed for comparison on each other and as such they don't at similar intensities.
Point 5: Line 203: Values of the vertical axis are missing;
Response 5: Here the graphs were superimposed, this is similar to response 4.
Point 6: Line 401: Units of the vertical axis are incorrect;
Response 6: The units are corrected.
Point 7: Line 427: Units of the vertical axis are wrong;
Response 7: The units are corrected.
Point 8: Line 427: Data in figure 10 need to be checked as it is over against data from figure 9.
Response 8: Figure 9 presents pure water at a continuous-time while Figure 10 presents antifouling behaviour at constant time intervals. In both figures, different membrane sections of similar membrane substrates were used, which amount to observing different water fluxes. This could be an indication that the attained membranes macrostructure is not uniform as exhibited by the presented micrographs.
Point 9: Line 433: Within which range the water flux was reduced ( need to show in %).
Response 9: Line 403-405-Rephrased the stamen to, ”this lower enhancement of water flux for F60-ZSM-22/PES than C60-ZSM-22 NF by about 150 L.m-2.h-1 can be explained by the difference in diffusion affinity of the resulting membranes and the inherent hydrophilic”.
Point 10: Line 450: Formulas for flux recovery ratios (FRR), total fouling (Rt), reversible fouling (Rr), and irreversible fouling (Rir) are needed.
Response 10: Formulas are included abd defined accordnily in section 2.4.
Point 11: Line 608: Units are incorrect.
Response 11: The units was corrected.

Round 2
Reviewer 2 Report
The authors covered the majority of points highlighted by the reviewers.